

# Degradation of zearalenone by microorganisms and enzymes

Jiregna Gari[1] and Rahma Abdella[2]

[1] Department of Veterinary Laboratory Technology, Ambo University, Ambo, Oromia, Ethiopia
[2] Armauer Hansen Research Institute, Addis Ababa, Ethiopia

## ABSTRACT

Mycotoxins are toxic metabolites produced by fungi that may cause serious health problems in humans and animals. Zearalenone is a secondary metabolite produced by fungi of the genus Fusarium, widely exists in animal feed and human food. One concern with the use of microbial strains and their enzyme derivatives for zearalenone degradation is the potential variability in the effectiveness of the degradation process. The efficiency of degradation may depend on various factors such as the type and concentration of zearalenone, the properties of the microbial strains and enzymes, and the environmental conditions. Therefore, it is important to carefully evaluate the efficacy of these methods under different conditions and ensure their reproducibility. Another important consideration is the safety and potential side effects of using microbial strains and enzymes for zearalenone degradation. It is necessary to evaluate the potential risks associated with the use of genetically modified microorganisms or recombinant enzymes, including their potential impact on the environment and non-target organisms. Additionally, it is important to ensure that the degradation products are indeed harmless and do not pose any health risks to humans or animals. Furthermore, while the use of microbial strains and enzymes may offer an environmentally friendly and cost-effective solution for zearalenone degradation, it is important to explore other methods such as physical or chemical treatments as well. These methods may offer complementary approaches for zearalenone detoxification, and their combination with microbial or enzyme-based methods may improve overall efficacy. Overall, the research on the biodegradation of zearalenone using microorganisms and enzyme derivatives is promising, but there are important considerations that need to be addressed to ensure the safety and effectiveness of these methods. Development of recombinant enzymes improves enzymatic detoxification of zearalenone to a non-toxic product without damaging the nutritional content. This review summarizes biodegradation of zearalenone using microorganisms and enzyme derivatives to nontoxic products. Further research is needed to fully evaluate the potential of these methods for mitigating the impact of mycotoxins in food and feed.

Corresponding author
Jiregna Gari, jiregna-gari2023@ambou.edu.et

## INTRODUCTION

Mycotoxins are naturally occurring toxic secondary metabolites of some microscopic filamentous fungi (*Liu, Xie & Wei, 2022*). Mycotoxins produced mainly by some fungal species belonging to *Alternaria*, *Aspergillus*, *Fusarium*, and *Penicillium* genera pose health

threats to humans and animals (*Greeff-Laubscher et al., 2020*). Mycotoxins contamination of foods and feeds is a current global issue and causes huge economic losses to animal husbandry (*Navale & Vamkudoth, 2022*). Zearalenone is an estrogenic mycotoxin produced by *Fusarium* species that leads to huge economic losses in the food industry and livestock husbandry. About 25–50% of the world's food crops are affected by mycotoxins (*Eskola et al., 2020*). The most economically important mycotoxins are aflatoxins, deoxynivalenol and zearalenone. Contamination of food and feed with zearalenone has reproductive problems, carcinogenicity, immunotoxicity, and other cytotoxic effects (*Ropejko & Twaruzek, 2021*; *Yli-Mattila et al., 2022*).

More than 400 different types of mycotoxins have been identified so far, with different levels of toxicity (*Arroyo-Manzanares et al., 2021*). Among mycotoxins, aflatoxins B1, zearalenone, ochratoxin A, patulin, and trichothecenes have received particular attention due to their severe health outcomes on both humans and animals, which can range from acute to severe and chronic intoxications in both humans and animals (*Ahn et al., 2022*; *Nahle et al., 2022*).

*Bouajila et al. (2022)* reported that zearalenone contaminate feeds like corn, wheat, barley, sorghum, rice have a variety of toxic effects on humans and animals (*Jia et al., 2022*). Zearalenone (ZEN) is a potent non-steroidal oestrogen mycotoxin which is biosynthesized *via* the polyketide pathway and could bind to estrogen receptors, which subsequently activate estrogen response elements in animals (*Singh & Kumari, 2022*; *Yli-Mattila et al., 2022*).

Zearalenone (ZEN) consumption causes hypoestrogenism in animals and interferes in the expression of estrogen and organ function (*Gajecka et al., 2021*). It could reduce the nutritional value of feed, damage the growth and health of livestock and poultry, and cause huge economic losses to livestock production. However, some animals, like chickens, show strong resistance to the toxicity of ZEN. ZEN can also cause abortion, infertility, stillbirth, and other reproductive effects on animals (*Yadav et al., 2021*; *Jia et al., 2022*).

In humans, ZEN has a chronic toxicity effect and stimulates the growth of mammary gland cells that might be involved in breast cancer (*Ropejko & Twaruzek, 2021*). There is a report that shows ZEN has immunotoxin, hepatotoxic, hematotoxic, and reproductive toxic effects like reducing fertility, vaginal prolapse, and causing vulvar swelling. The two primary metabolites of zearalenone are $\alpha$-zearalenol ($\alpha$-ZEL), which is a synthetic version of zearalenone, and $\beta$-zearalenol ($\beta$-ZEL), which is produced by reducing ZEN. Zearalenone is metabolized in the intestinal cells. Zearalenone also comes in the forms of $\alpha$-zearalanol ($\alpha$-ZAL) and $\beta$-zearalanol ($\beta$-ZAL). It is capable of being conjugated with glucuronic acid in its metabolized state (*Ropejko & Twaruzek, 2021*). The degradation of zearalenone toxicity is commonly done by the use of physical, chemical, and biological approaches. Zearalenone is heat-stable and shows great resistance to conventional degradation methods (*Kabak, Dobson & Var, 2006*; *Wu et al., 2021*). However, physical and chemical degradation destroys nutritional structure, decreases palatability of the feed and causes pollution to the environment (*Guan et al., 2021*). Biological degradation has great specificity and degrades zearalenone completely without producing harmless products (*Xu et al., 2022*).

At present, microorganisms and enzymes derived from microbial strains have been widely used for the degradation of zearalenone in food and feed (*Nahle et al., 2022*; *Xu et al., 2022*). Researchers have developed biodegradation of zearalenone by the use of microbial and their enzyme derivatives, which offers harmless products and is environmentally friendly. Recently, numerous studies have focused on degradation through biological approaches by using microorganisms including bacteria, yeast, and fungi, and microorganisms' enzymes to remove zearalenone from food sources (*Luo et al., 2020*; *Nahle et al., 2022*). Development of genetic engineering technology in the advancement of recombinant proteins improves enzymatic degradation of zearalenone (*Guan et al., 2021*). This review aims to discuss the biological degradation of ZEN through microorganisms and enzymes developed in recent years.

## SURVEY METHODOLOGY

The varieties of mycotoxin-degrading microorganisms and enzymes, the development of heterologously generated degrading enzymes through genetic engineering, and related studies on enhancing the efficacy of degrading enzymes were all summarized in this review. The published articles were gathered using the databases Science Direct, Scopus, PubMed Web of Science, and a Gray literature resource like Google Scholar. The following keywords were used to search for the review: [Zearalenone Degradation OR Microorganisms OR Enzyme] and [zearalenone]. After passing the abstract screening, the full text of the found publications was downloaded. Any manuscript that wasn't available in this regard was discarded. For data extraction and analysis, only the articles with accessible full texts underwent further screening.

### Degradation of zearalenone by microorganisms

Microbial degradation occurs when microorganisms (bacterial and yeast) secrete their metabolites or enzymes during their growth and development process. Microorganisms can directly adsorb targeted toxins or reduce toxins of our interest to impede the production of mycotoxins (*Feng et al., 2020*; *Xu et al., 2022*). Many studies have reported on the biodegradation of ZEN using microorganisms (Table 1). They show high specificity and eco-friendliness in decreasing the possibility of ZEN toxicity from food and feed (*Song et al., 2021*).

A variety of non-pathogenic microbes like probiotics, *Bacillus*, *Saccharomyces*, and *Lactobacillus* species have a high capability to detoxify feeds contaminated with zearalenone because they follow standards like safe to be used and possess detoxifying ability without forming bad odor or taste in the feeds (*Wang et al., 2019*; *Zhu et al., 2021*). Several studies reveal detoxification of zearalenone using probiotics, including by yeast, *Bacillus,* and lactic acid bacteria (Table 1) as they are involved in adsorption of ZEN and preventing its absorption by animals (*Hathout & Aly, 2014*).

Various bacteria, yeasts, and fungi can convert structure of ZEN to alpha and beta zearalenol through hydrolysis, conjugation of sulfate and glucosyl group reduction (*Cho et al., 2010*). Among *Bacillus* strains, *B. licheniformis, B. subtilis, B. natto, and B. cerues* were those found to have the highest detoxification effect on zearalenon in food and feed (*Wang*

**Table 1** Recent research that shows microorganisms used for the degradation of zearalenone (ZEN).

| Food source or media used | Strain | ZEN concentration | Degradation range | References |
|---|---|---|---|---|
| Liquid LB medium | *Streptomyces rimosus* (K145, K189) | 1 μg mL−1 | 100% | *Harkai et al. (2016)* |
| Feed | *Bacillus licheniformis* CK1 | 1.20 ± 0.11, 0.47 ± 0.22 mg/kg | Can degrade ZEN | *Fu et al. (2016)* |
| Liquid chromatography-tandem mass spectrometry and Thin layer chromatography | *Candida parapsilosis* | 20 μg/mL | Decreased by 97% | *Pan et al. (2022)* |
| Potassium phosphate buffer | *Lact. plantarum* 3QB361 | 2 μg/mL | 82% | *Møller et al. (2021)* |
| Aqueous solution | *Lact. plantarum* BCC 47723 | 0.2 μg/mL | 0.5%–23% | *Adunphatcharaphon, Petchkongkaew & Visessanguan (2021)* |
| Culture medium/liquid food/solid-state fermentation | *Bacillus subtilis* *Bacillus natto* | 20 ug/mL; 1 mg/kg; 20 μg/mL | 100% and 87% 65, 73%/75%, 70% | *Ju et al. (2019)* |
| Nutrient broth | *Bacillus subtilis, Candida utilis, Aspergillus oryzae* | 1 μg/mL | 92.27% *A. oryzae.* combined form can degrade 95.15% | *Liu et al. (2019)* |
| Malting wheat grains with bacterial suspension | *P. acidilactici* | 19.5–873.7 μg/L | 38.0% | *Juodeikiene et al. (2018)* |
| LB medium and simulated gastric fluid (GSF) | *Bacillus cereus* BC7 | 10 mg/L | 100% and 89.31% | *Wang et al. (2018)* |
| Corn meal medium | *B. licheniformis* CK1 | 5 μg/mL | 73% | *Hsu et al. (2018)* |
| Culture medium | *Bacillus pumilus* ES 21 | 17.9 mg/ml | 95.7% | *Wang et al. (2017)* |
| MRS broth | *Lactobacillus rhamnosus* | 200 μg/mL | Showed the highest adsorption (68.2%) | *Vega et al. (2017)* |
| MRS broth | *Lactobacillus plantarum* ZJ316 | 5 mg/L | highest ZEA degradation ability | *Chen et al. (2018)* |
| The LB medium | *Acinetobacter calcoaceticus* | 5 μg/mL | 85.77% | *Deng et al. (2021)* |

**Table 1** (*continued*)

| Food source or media used | Strain | ZEN concentration | Degradation range | References |
|---|---|---|---|---|
| HPLC-TOF-MS and NMR | *B. subtilis* Y816 | 40 mg/L | Transform of ZEN within 7 h | *Bin et al. (2021)* |
| Cell suspensions on MRS agar | *Lb.fermentum* 2I3, *Lb.reuteri* L26, *Lb.plantarum* L81, *Lb.reuteri*, *Lb.plantarum* CCM 1904, | 0.01 ppm | (57.9—100)% | *Harčárová et al. (2022)* |
| Cell suspensions on MRS agar | *Bacillus subtilis* CCM 2794 | 0.01 ppm | 11.7% | *Harčárová et al. (2022)* |

*et al., 2019*). *Bacillus pumlius* ANSB01G is also reported to degrade ZEN in the feed of animals (*Xu et al., 2022*). According to *Xu et al. (2016)*, *B. amyloliquefaciens* ZDS-1 has ZEN degrading ability in screened colonies. Probiotics is a great choice for biodegradation of ZEN in the food industry because it shows health benefits for humans and animals. Most lactic acid bacteria (LABs) are considered safe probiotics in the food industry. It is reported that *Lactobacillus* strains have a potential role in degrading ZEN from fermented food products (*Średnicka et al., 2021*). *Lact. paracasei*, and *Lc. lacti* have the ability to remove ZEN in aqueous food solutions (*Wu et al., 2021*) . There is a report that shows zearalenone can be degraded from PBS buffer solution by *Lact. Acidophilus* CIP 76.13T by a bioremediation range of 57% (*Ragoubi et al., 2021*).

There is a report that shows *B. licheniformis* CK1 has good probiotic properties and can degrade ZEN by more than 90% after 36 h of incubation in the contaminated corn meal medium by ZEN (*Hsu et al., 2018*). Other strains of fungi called *Saccharomyces cerevisiae* also have high ZEN degradation abilities. There is a report that shows *S. cerevisiae* isolate from grape can degrade ZEN (*Rogowska et al., 2019*). *Saccharomyces cerevisiae* isolated from silage has biodegradation properties and can degrade up to 90% of ZEN in two days (*Keller et al., 2015*). According to *Harkai et al. (2016)*, the bacteria *Streptomyces rimosus* (K145, K189) can degrade ZEN in liquid media. *Wang et al. (2018)* also investigated a *Lysinibacillus* strain isolated from chicken's large intestine digesta is capable of degrading zearalenone. Diagrammatic pathway for the process of degradation of zearalenone by microorganism (Fig. 1) source (*Mukherjee et al., 2014*).

## Degradation of zearalenone by enzymes

Recent advancements in genetic engineering technology have attracted researchers' attention towards recombinant enzymes to degrade mycotoxins in food and feed with high efficiency. The attainment and cloning of recombinant enzyme genes leads to the safe expression of genes in microbes, which has become a novel progress in molecular modification for ZEN degradation (*Azam et al., 2019*; *Xu et al., 2022*). Enzymatic degradation has wide advantages over microbial degradation because it can perform biodegradation with high efficiency, lower cost, reproducibility, and homogenous performance (*Loi et al., 2017*; *Liu, Xie & Wei, 2022*).

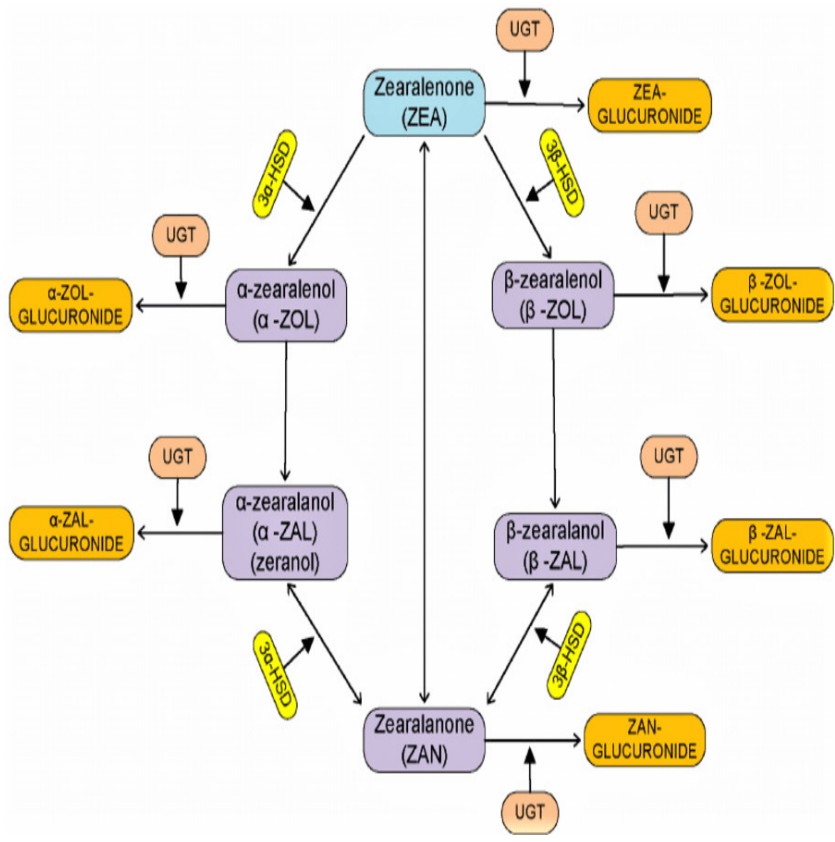

**Figure 1** Diagrammatic pathway for the process of degradation of zearalenone by microorganism.

A bacterial strain of *E. coli*, *S. cerevisiae*, and *Pichia pastoris* has been reported to remove ZEN from culture medium (*Wang & Xie, 2020*). *Gao et al. (2022)* identify and describe the activity of the ZEN degrading enzyme from *Exophiala spinifera*, ZHD_LD. Recently, microbial strains that are able to degrade ZEN have been isolated, and subsequently genes like ZHD101, ZLHY-6, and ZEN-jjm, as well as ZHD518 have been cloned (*Cheng et al., 2010*). ZHD101 is one of the recombinant enzymes derived from *Clonostachys rosea* that degrades ZEN (*Yang et al., 2017*).

*Wang et al. (2018)* reported that the lactonohydrolase Zhd518 enzyme in *E. coli* has high biodegrading ability against ZEN in food and feed industries. There is a study that shows RmZHD, a ZEN hydrolyzing enzyme from *Rhinocladiella mackenziei*, has the ability to degrade ZEN (*Zhou et al., 2020*). Recombinant Prx (peroxiredoxin), a cloned gene from *Acinetobacter* sp. SM04 expressed in *E. coli*, has the ability to degrade ZEN in the presence of hydrogen peroxide (*Yu et al., 2012*). It has been reported that laccase enzymes that are found on bacterial and yeast cells have the ability to degrade mycotoxins (*Guo et al., 2020*). *Song et al. (2021)* show the laccase gene obtained from the fungus *P. pulmonarius* has an enzymatic property to degrade zearalenone when it was expressed in the *Pichia pastoris* X33 yeast strain by producing recombinant protein.

**Table 2  Enzymatic degradation of zearalenone (ZEN).**

| Enzymes name | Source | Expression System | Degrading properties | References |
|---|---|---|---|---|
| Peroxiredoxin | *Acinetobacter* sp. SM04 | *S. cerevisiae* | Optimal activity at pH 9.0, 80 °C and $H_2O_2$ concentration of 20 mmol/L Thermal stable, alkali resistance | *Tang et al. (2013)* |
| Lactone hydrolase ZHD | *Gliocladium roseum* | *P. pastoris* | Enzyme activity in flask fermentation was 22.5 U/mL and specific activity of 4976.5 U/mg. Maximum enzyme activity of the supernatant was 150.1 U/ml in 5-L fermenter | *Xiang et al. (2016)* |
| Cb ZHD | *C. rosea* | *Cladophialophora bantiana* | Optimal enzyme activity at temperature 35 °C and pH 8 | *Hui et al. (2017)* |
| Lactonohydrolase | *Clonostachys rosea* | *Lactobacillus reuteri Pg4* | Not affect cell growth, acid and bile salt tolerance | *Yang et al. (2017)* |
| Lactonohydrolase Zhd518 | *Clonostachys rosea* | *E. coli* | Activity of 207.0 U/mg with optimal temperature 40 °C and pH 8. | *Wang et al. (2018)* |
| Lactonase | *Neurospora crassa* | *P. pastoris* | Optimal activity at pH 8.0 and 45 °C, stable at pH 6.0–8.0 for 1 h at 37 °C, Maximal enzyme activity at 290.6 U/mL in 30-L fermenter | *Bi et al. (2018)* |
| Lactonehydrolase ZENC | *Neurospora crassa* | *P. pastoris* | 99.75% of ZEN (20 μg/ml) was degraded at pH 8.0, 45 °C for 15 min | *Bi et al. (2018)* |
| Fusion ZHDCP enzyme | *C. rosea* *B.amyloliquefaciens* ASAG | *E. coli* | 100% degradation rate at pH 7 and 30 °C | *Azam et al. (2019)* |
| ZLHY-6 | *Pichia pastoris* | *P. pastoris* GSZ | low nutrient loss safe removal of ZEN | *Chang et al. (2020)* |
| *lac2* | *Pleurotus pulmonarius* | *P. pastoris* X33 | Lac2-ABTS and Lac2-AS degrade ZEN at optimum pH 3.5 and temperature 55 °C of recombinant *Lac2* | *Song et al. (2021)* |
| Lactonohydrolase | *Trichoderma aggressivum* | *E. coli* BL21 | With superior pH stability, the surface exhibit ZHD-P retained 80% activity | *Chen et al. (2021)* |
| ZPF1 | *C. rosea* fused with *Phanerochaete chysosporium* | *Kluyveromyces lactis* GG799 | ZEN degraded up to 46.21% ± 3.17% | *Xia et al. (2021)* |
| DyP | *Streptomyces thermocarboxydus* 41291 | *E. coli* BL21 | ZEN was degraded slightly by StDyP | *Qin et al. (2021)* |
| Ase | *Acinetobacter* Sp | *E. coli* BL21 | Degraded 88.4% of ZEN (20 μg/mL) | *Tang et al. (2022)* |

Studies have shown that laccase enzymes are considered to be an effective zealenone toxicity antidote. Furthermore, *Pleurotus eryngii* laccase enzyme can degrade aflatoxin B1, ochratoxin A, zearalenon, and other mycotoxins (*Wu et al., 2021*). A gene ZENC, zearalenone lactonase gene from *Neurospora crassa*, is expressed in *P. pastoris*. It had a maximal enzyme activity when fermented using high density fermatation at pH 8 and a temperature of 45 °C. Furthermore, ZENC was also found to be effective in ZEN containing feed materials with a high degradation rate (*Bi et al., 2018*).

*Garcia, Feltrin & Garda-Buffon (2018)* also reported that the peroxidase enzyme has the ability to degrade zearalenone concentrations. According to the study, fusion of multifunctional recombinant enzymes ZHDCP with genes of ZEN hydrolases and carboxypeptidases has the ability to detoxify zearalenone in 2 h at pH and temperature of 35 °C (*Azam et al., 2019*). Many studies shows that enzymes can able to degrade zearalenone as expressed Table 2 (Table 2).

## CONCLUSIONS

The severe impact of zealarenone on animals and humans' health, present in contaminated food and feed, has received global attention. Many approaches have been established for the removal of ZEN. Biodegradation is considered the safest approach because it degrades toxins without residual toxic substances. Recent research shows the development of recombinant microorganisms and recombinant enzymes to detoxify ZEN in foods and feeds. However, the health impacts of recombinant enzymes are not clearly described. Currently, biodegradation of zealarenone is laboratory-based. The commercial scale of biodegradation needs further studies. Further interdisciplinary studies concerning gene cloning, genetic modification of microorganisms, and the development of recombinant enzymes are promising approaches for safe zealarenone degradation. Future study should pay particular attention to the effects of toxin levels close to those experienced by humans, the choice of animal models, and the application of pathogenic mechanisms that differ greatly from humans. The emergence of microbial and enzyme preparations is quickly approaching the point at which it can be industrialized. The promise of these techniques for lessening the effects of mycotoxins in food and feed still need more study.

### Funding
The authors received no funding for this work.

### Competing Interests
The authors declare there are no competing interests.

### Author Contributions
- Jiregna Gari conceived and designed the experiments, performed the experiments, analyzed the data, prepared figures and/or tables, authored or reviewed drafts of the article, and approved the final draft.

- Rahma Abdella conceived and designed the experiments, analyzed the data, authored or reviewed drafts of the article, and approved the final draft.

## Data Availability

This is a literature review.

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
