# Peer review of "Degradation of zearalenone by microorganisms and enzymes"

_PeerJ, doi:10.7717/peerj.15808_

## Round 0.1 · original submission · Major Revisions

Additional comments:
One concern with the use of microbial strains and their enzyme derivatives for zearalenone degradation is the potential variability in the effectiveness of the degradation process. The efficiency of degradation may depend on various factors such as the type and concentration of zearalenone, the properties of the microbial strains and enzymes, and the environmental conditions. Therefore, it is important to carefully evaluate the efficacy of these methods under different conditions and ensure their reproducibility.

Another important consideration is the safety and potential side effects of using microbial strains and enzymes for zearalenone degradation. It is necessary to evaluate the potential risks associated with the use of genetically modified microorganisms or recombinant enzymes, including their potential impact on the environment and non-target organisms. Additionally, it is important to ensure that the degradation products are indeed harmless and do not pose any health risks to humans or animals.

Furthermore, while the use of microbial strains and enzymes may offer an environmentally friendly and cost-effective solution for zearalenone degradation, it is important to explore other methods such as physical or chemical treatments as well. These methods may offer complementary approaches for zearalenone detoxification, and their combination with microbial or enzyme-based methods may improve overall efficacy.

Overall, the research on the biodegradation of zearalenone using microorganisms and enzyme derivatives is promising, but there are important considerations that need to be addressed to ensure the safety and effectiveness of these methods. Further research is needed to fully evaluate the potential of these methods for mitigating the impact of mycotoxins in food and feed.

Reviewer 1 ·

Basic reporting

After reading this review article, it seems like there are a few places where it may be improved. In particular, the work lacks novelty and provides little more knowledge or insights to what has previously been published in the body of existing research. Also, to make the paper more interesting and readable for the target audience, the writing style might be improved. The authors should think about including more original research or discoveries or presenting the data in a more creative or interesting manner.

The abstract could benefit from more specific information about the sources of the research that it summarizes. The abstract does not provide any information on the specific microorganisms and enzymes that have been documented for the degradation of zearalenone. This lack of detail may make it difficult for readers to fully understand the implications of the findings.

Experimental design

The section "Survey methodology" is not in consensus with the actual findings and documentation.

Validity of the findings

More emphasis should be given towards the mechanistic understanding of microbes and enzymes. Some good flow charts and illustrative models will add value to this work.

Reviewer 2 ·

Basic reporting

The review undertaken is of importance to economic sustainability to foods and feeds. The sequential flow is jumbled and should be aligned conceptually. The language is standard and should be improved for journal standards. Word strength including molecular and mechanistic approach should be improved in the review.
Comments to authors

The review titled “Degradation of zearalenone by microorganisms and enzymes” deals with the negative impact of the mycotoxin zearalenone on animal and plant health and its future remedies. The review article is well constructed, will be of scientific interest and valuable for future agriculturist and veterinarians to get insights into the above area. In my opinion, this review article lacks deep insight explanation overall. The article information seems repetitive, requires more information and lacks sequential flow. The overall word strength of the article can be worked upon and more fluent language of vocabulary can be served.

I find minor typos, grammatical errors and jumbled vocabulary throughout the manuscript that makes comprehension difficult. Kindly simplify for constructive flow of article concept for better insight into the topic. The informative lines seem cyclic and no mechanistic insight is provided in the article. Therefore, I would suggest the authors to carry out a careful and extensive revision of the text and include the required mentioned data to make the article more significant and impactful.

Title: Kindly recast the title hinting the destructive importance of zearalenone.
Abstract: Abstract should be informative fulfilling required data on economic loss and increase the word count.
Introduction: Restructure the introduction part with a conceptual understanding and include more references pertaining to fungus, mycotoxins. Include alternate synonyms of a repeated word.
Review: Lacks molecular mechanisms information’s that controls or modulate the pathogenicity of various mycotoxins by microorganisms and enzymes. Include future thrusts.






Section Line no Comments
Abstract 15 Add economic loss data.
17 ‘other cytotoxic effects’ kindly elaborate
18 In the manuscript there is repetitive use of ‘food and feed’. Kindly add references for food affected by mycotoxins.
Introduction 28 Kindly recast with corrected grammar.
32 Quantify the economic losses caused by fungal mycotoxins with references.
36 Kindly recast with corrected grammar.
42 ‘other grains’ kindly elaborate the information.
56 Kindly provide the mechanism involved in such occurrence. Factors governing such outcomes.
58 Describe the various physical and chemical methods in biodegradation.
99 Alpha and beta zearanalenol, kindly explain the derivative pathway.
120 Sentence incomplete.
132 Kindly provide full scientific names of previously not mentioned organisms.
Conclusion Kindly recast the conclusion with future thrusts.

Experimental design

The design followed is accepted.

Validity of the findings

No Comment

Additional comments

No comment.

---

## Round 0.2 · Major Revisions

Authors are requested to resolve the query raised by the reviewers.

Reviewer 1 ·

Basic reporting

All the queries were addressed.

Experimental design

Design is appropriate

Validity of the findings

The findings are accurate.

·

Basic reporting

Sir,
I find the work 'Degradation of zearalenone by microorganisms and enzymes' interesting and congratulate the authors for carrying out this intensive review. While reviewing the MS some comments have been given to improve the quality of MS also to improve its effectiveness with respect to readers. The comments have been given in the annotated PDF file.

Submitting for needful.

Experimental design

Not applicable, However on diagrammatic representation has been suggested for improving the quality of MS.

Validity of the findings

Satisfactory

Additional comments

The MS needs incorporation of some important points before its acceptance for publication.

---

## Round 0.3 · accepted · Accept

The authors have addressed all of the reviewers' comments. Hence, the MS can be accepted in its current form.

Reviewer 2 ·

Basic reporting

The authors have revised the manuscript according to the observations and now it is acceptable.

Experimental design

Design is appropriate

Validity of the findings

NA

Additional comments

NA

·

Basic reporting

The authors have incorporated the suggested changes, the article may be accepted now for possible publication in PeerJ.

Experimental design

Satisfactory

Validity of the findings

Satisfactory

Additional comments

NA